# Disialoganglioside GD2-Targeted Near-Infrared Photoimmunotherapy (NIR-PIT) in Tumors of Neuroectodermal Origin

**DOI:** 10.3390/pharmaceutics14102037

**Published:** 2022-09-24

**Authors:** Fuyuki F. Inagaki, Takuya Kato, Aki Furusawa, Ryuhei Okada, Hiroaki Wakiyama, Hideyuki Furumoto, Shuhei Okuyama, Peter L. Choyke, Hisataka Kobayashi

**Affiliations:** Molecular Imaging Branch, Center for Cancer Research, National Cancer Institute, National Institutes of Health, Bethesda, MD 20892, USA

**Keywords:** glycosphingolipid, ganglioside, immunotherapy, neuroblastoma, IR700

## Abstract

Disialoganglioside (GD2) is a subtype of glycolipids that is highly expressed in tumors of neuroectodermal origins, such as neuroblastoma and osteosarcoma. Its limited expression in normal tissues makes GD2 a potential target for precision therapy. Several anti-GD2 monoclonal antibodies are currently in clinical use and have had moderate success. Near-infrared photoimmunotherapy (NIR-PIT) is a cancer therapy that arms antibodies with IRDye700DX (IR700) and then exposes this antibody–dye conjugate (ADC) to NIR light at a wavelength of 690 nm. NIR light irradiation induces a profound photochemical response in IR700, resulting in protein aggregates that lead to cell membrane damage and death. In this study, we examined the feasibility of GD2-targeted NIR-PIT. Although GD2, like other glycolipids, is only located in the outer leaflet of the cell membrane, the aggregates formation exerted sufficient physical force to disrupt the cell membrane and kill target cells in vitro. In in vivo studies, tumor growth was significantly inhibited after GD2-targeted NIR-PIT, resulting in prolonged survival. Following GD2-targeted NIR-PIT, activation of host immunity was observed. In conclusion, GD2-targeted NIR-PIT was similarly effective to the conventional protein-targeted NIR-PIT. This study demonstrates that membrane glycolipid can be a new target of NIR-PIT.

## 1. Introduction

Gangliosides are sialic acid-containing glycosphingolipids that play important roles in signal transduction and cell adhesion. While most ganglioside subtypes are widely expressed in normal tissues, disialoganglioside (GD2) is rarely expressed in normal tissues but is selectively expressed in neuroectodermal origin tumors such as neuroblastoma, osteosarcoma, Ewing sarcoma, soft tissue sarcoma, melanoma, lymphoma, and small-cell lung carcinoma [1,2,3,4,5]. Thus, GD2 is considered a tumor-associated antigen that can direct tumor-targeted therapies. In tumors, the presence of GD2 is associated with more malignant phenotypes characterized by increased cell proliferation, migration, invasion, and immune evasion, depending on the tumor type [6,7,8]. However, the mechanisms by which these occur are not well understood. Because of its limited expression in normal tissues (other than the central nervous system, peripheral nerves, and melanocytes), GD2 is an attractive target for molecular targeting.

Several antibodies against GD2 have been tested in clinical trials and used in neuroblastoma. Dinutuximab (ch14.18) is the first anti-GD2 monoclonal antibody drug, which was approved by the Europeans Medicines Agency (EMA) and the US Food and Drug Administration (FDA) for the treatment of high-risk pediatric neuroblastoma. This drug was tested in combination with interleukin-2 (IL-2), granulocyte-macrophage colony-stimulating factor (GM-CSF), and 13-cis retinoic acid (RA) [9]. Dinutuximab beta (ch14.18/CHO) was subsequently approved by EMA [10]. These antibodies have been successful in clinical trials and are now part of the standard treatments for neuroblastoma. Still, nearly half of patients experience recurrence, and 20% of patients do not respond to initial induction therapy with anti-GD2 monoclonal antibodies. In addition, anti-GD2 monoclonal antibody therapy against other GD2 positive tumors, including osteosarcoma and small cell lung carcinoma, is commonly ineffective [11,12]. Therefore, only modest effects are seen with anti-GD2 monoclonal antibody therapy [13].

Near-infrared photoimmunotherapy (NIR-PIT) is a newly developed cancer treatment which utilizes an intravenously administered antibody-IRDye700DX (IR700) conjugate that binds to the tumor. Subsequent exposure to NIR light at 690 nm [14,15,16] results in selective cancer cell death. NIR light causes profound photochemical changes in the antibody–dye conjugate (ADC), resulting in dramatically increased hydrophobicity that results in physical damage to the cell membrane. Within minutes, targeted cells rupture, inducing selective necrotic cell death [17]. In pre-clinical models, NIR-PIT can be successfully used with a variety of different antibodies conjugated to IR700 [18,19,20,21]. A phase 1/2a, open-label, multicenter clinical study using Epidermal growth factor receptor (EGFR)-targeted cetuximab-IR700 (RM-1929) in patients with locoregional recurrent head and neck cancer demonstrated good tolerability and clinically meaningful response and survival [22]. The results of this study, along with positive results from another phase 1/2a clinical trial [23], led to the qualified approval of the first EGFR-targeting NIR-PIT drug (Akalux^®^, Rakuten Medical Inc., San Diego, CA, USA) and a 690 nm laser system (BioBlade^®^, Rakuten Medical Inc.) from the Japanese Ministry of Health, Labor and Welfare in September 2020. FDA-designated fast-track global Phase 3 clinical trial is underway (https://clinicaltrials.gov/ct2/show/NCT03769506 (accessed on 21 September 2022)).

In this study, we determine if anti-GD2 antibody-IR700 is a suitable ADC for NIR-PIT based on in vitro and in vivo experiments with three GD2-expressing tumor cell lines: EL4-luc (murine lymphoma), LAN-1 (human neuroblastoma), and T98G (human glioblastoma).

## 2. Materials and Methods

### 2.1. Reagents

Water-soluble, silica-phthalocyanine-derivative, IRDye700DX NHS ester (IR700) was purchased from LI-COR Bioscience (Lincoln, NE, USA). Anti-human Ganglioside GD2 monoclonal antibody (clone 14G2a) was purchased from Bio X Cell (Lebanon, NH, USA). All other chemicals were of reagent grade.

### 2.2. Antibody-IR700 Conjugation

Conjugation of IR700 to anti-GD2 monoclonal antibody was performed as previously described [24]. Briefly, anti-human Ganglioside GD2 antibody (1 mg, 6.8 nmol) was mixed with IR700 (34.2 nmol, 10 mmol/L in DMSO) in 0.1 mol/L Na_2_HPO_4_ (pH 8.5). The mixture was reacted at room temperature for 60 min. Then, the reactants were purified using a gel filtration column (Sephadex G 25 column, PD-10, GE Healthcare, Piscataway, NJ, USA). IR700 conjugated to anti-GD2 (αGD2) antibody is abbreviated as αGD2-IR700. The number of IR700 bound to each antibody was evaluated by measuring the absorbance at 280 nm and 689 nm using ultraviolet-visible spectroscopy (8453 Value System; Agilent Technologies, Santa Clara, CA, USA).

### 2.3. Sodium Dodecyl Sulfate Polyacrylamide Gel Electrophoresis (SDS-PAGE)

αGD2 antibody and αGD2-IR700 were analyzed with a 4–20% gradient polyacrylamide gel (Life Technologies, Gaithersburg, MD, USA) using our standard procedures reported in previous studies [21].

### 2.4. Size-Exclusion Chromatography

Size-exclusion chromatography (SEC) analysis was performed using TSKgel SuperSW 3000 (4.6 mm × 30 cm, 5 μm) with a guard column (Tosoh Bioscience, Inc., South San Francisco, CA, USA) equipped on the Nexera XR UHPLC system (Shimadzu Co., Kyoto, Japan) as previously reported as the standard procedures [21].

### 2.5. Cell Culture

A human neuroblastoma cell line, LAN-1, was purchased from ECACC. A human glioblastoma cell line, T98G, and a murine lymphoma cell line, EL4-luc, were purchased from ATCC. The EL4-luc cell line is the EL4 cell line transfected with the firefly luciferase gene and stably expressing luciferase. LAN-1 cells were cultured in EMEM (with non-essential amino acids) and Ham’s F12 (1:1 mixture) supplemented with 10% FBS, 2 mM Glutamine, and penicillin (100 units/mL)/streptomycin (100 µg/mL) (GIBCO, Waltham, MA, USA) under the standard conditions as previously described [21].

### 2.6. Flow Cytometric Analysis of GD2 Expression

GD2 expression on each cell line was measured using a flow cytometer (FACS Lyrics, BD Biosciences, San Jose, CA, USA) and analyzed with FlowJo software (Version 10.6.2; BD Biosciences, Franklin Lakes, NJ, USA). One million cells were mixed with 1 μg of αGD2-IR700 and reacted for 2 h on ice. To confirm the specific binding of the conjugated antibody, 50 μg of excess unconjugated antibody were added to block the conjugated antibody.

### 2.7. Fluorescence Microscopic Studies

To observe the morphological change in cells after GD2-targeted NIR-PIT, fluorescence microscopy was performed (BX61; Olympus America Inc., Center Valley, PA, USA). Five thousand EL4-luc, LAN-1, or T98G cells were seeded on a 6-well plate with a cover glass and cultured for 24 h. αGD2-IR700 was then added to the culture medium at a final concentration of 10 μg/mL. After 3 h of incubation at 37 ℃, cells were washed with PBS. The cells were then irradiated with a NIR-laser (690 ± 5 nm, 150 mW/cm^2^) at a density of 16 J/cm^2^, using an ML7710 laser system (Modulight Inc, Tampere, Finland). Transmitted light differential interference contrast (DIC) images were acquired before and after NIR-PIT. Dead cells were stained with propidium iodide (PI). A filter set consisting of an excitation filter from 532.5 to 587.5 nm and a band-pass emission filter from 607.5 to 682.5 nm was used to detect PI fluorescence.

### 2.8. In Vitro NIR-PIT

The cytotoxic effects of GD2-targeted NIR-PIT were determined by flow cytometry. A half million cells in PBS were dispensed into each tube, and then 10 μg/mL of αGD2-IR700 was added. After 1 h of incubation at 4 ℃, cells were washed with PBS, and NIR-laser light (690 ± 5 nm, 150 mW/cm^2^) was applied at indicated energy densities (ML7710 laser system). Cells were collected after 30 min of incubation, and PI was added. Cells were analyzed by FACS Lyrics and FlowJo Software.

### 2.9. Animals and Tumor Models

All animal procedures were performed in compliance with the Guide for the Care and Use of Laboratory Animal Resources (1996) and the National Research Council, and approved by the local Animal Care and Use Committee. Six-week-old female C57BL/6 mice (strain #000664) were purchased from the Jackson Laboratory. Mice were maintained under a 12 h light-dark cycle. After a 7–10 day acclimation period, mice were anesthetized with 2.5–4.0% inhaled isoflurane and/or intraperitoneal injection of sodium pentobarbital (Nembutal Sodium Solution, Ovation Pharmaceuticals Inc., Deerfield, IL, USA). In the subcutaneous tumor model, EL4-luc cells were subcutaneously injected into the right lower flank. Mouse fur at and around the tumor site was shaved prior to imaging studies and NIR-PIT experiments. Tumor volume was determined by measuring the greatest longitudinal diameter (length) and the greatest transverse diameter (width) with a caliper. Tumor volume was calculated as follows: tumor volume = length × width^2^ × 0.5. Tumor size was measured daily until the tumor volume reached 2000 mm^3^ or the length reached 2 cm, whereupon the mice were euthanized with inhalation of carbon dioxide.

### 2.10. Biodistribution Study

EL4-luc cells (1 × 10^5^) were subcutaneously injected into the right lower flank of mice. Six days after cell inoculation, fluorescence imaging studies were performed. Serial ventral and dorsal fluorescence images were obtained with Pearl Imager (LI-COR Biosciences, Lincoln, NE, USA) using an IR700 fluorescence channel (Ex; 685 nm, Em; 720 nm) before and 1, 2, 3, 6, 9, 12, 24, 36, 48, 72, 96, 120, and 144 h after intravenous injection (i.v.) of 100 μg of αGD2-IR700. Regions of interest (ROIs) were placed on the tumor and liver. Background ROIs were also placed in non-target regions. The average fluorescence intensity of each ROI and target-to-background ratios (TBR = fluorescence intensities of target/fluorescence intensities of background) were calculated using Pearl Cam Software (LI-COR Biosciences).

### 2.11. In Vivo NIR-PIT

To assess the in vivo therapeutic efficacy of GD2-targeted NIR-PIT, thirty EL4-luc tumor-bearing mice were randomly divided into the following three groups: (i) no treatment (Ctrl group); (ii) 100 μg of αGD2-IR700 i.v. without NIR light exposure (ADC i.v. group); (iii) 100 μg of αGD2-IR700 i.v. with NIR light exposure (NIR-PIT group). Five days after tumor cell inoculation, 100 μg of αGD2-IR700 was administered via tail vein. NIR light laser (ML7710 laser system; 690 ± 5 nm, 150 mW/cm^2^, 50 J/cm^2^) was exposed the next day. Fluorescence images of IR700 were acquired pre-and post-NIR-PIT using 700 nm fluorescence channel of the Pearl Imager.

### 2.12. Bioluminescence Imaging

Bioluminescence imaging (BLI) was performed by intraperitoneally injecting D-Luciferin (15 mg/mL, 200 μL) (Gold Biotechnology, St. Louis, MO, USA). Images were obtained and analyzed with the standard procedures as previously described [18].

### 2.13. Hematoxylin and Eosin Stain (H&E Stain)

One day after NIR-PIT, tumors were excised for histological studies. Resected tumors were fixed with 10% neutral buffered formaldehyde and embedded in paraffin. Tissue sections were stained with hematoxylin and eosin. Pathologic findings were investigated using our standard procedures [18].

### 2.14. Flow Cytometric Analysis of Regional Lymph Node

One day after NIR-PIT, tumor-draining lymph nodes were removed. Single-cell suspension was prepared from the resected lymph nodes by mechanical crushing and filtration (70 μm cell strainer). The cells were stained with the following monoclonal antibodies: anti-CD45 (clone 30-F11, Thermo Fisher Scientific, Waltham, MA, USA), anti-CD3ε (clone 145-2C11, BioLegend, San Diego, CA, USA), anti-CD8α (clone 53–6.7, Thermo Fisher Scientific), anti-CD25 (clone PC61, BioLegend), anti-CD69 (clone H1.2F3, Thermo Fisher Scientific), and anti-NK1.1 (clone PK136, BioLegend). The cells were also stained with Fixable Viability Dye (Thermo Fisher Scientific) in order to exclude the dead cells from the study. The samples were then analyzed with a cell analyzer (FACS Lyrics and FlowJo software). Each cell population was determined as follows: killer T cell = CD45+/CD3+/CD8+; NK cell = CD45+/CD3−/NK1.1+; NKT cell = CD45+/CD3+/NK1.1+.

### 2.15. Statistical Analysis

Statistical analysis was conducted with GraphPad Prism version 8 (GraphPad Software, La Jolla, CA, USA). A one-way analysis of variance (ANOVA) followed by Tukey’s test was performed to compare multiple groups based on a single measurement. A two-way repeated measures ANOVA followed by Tukey’s test was performed to compare luciferase activity and tumor volumes. Data were presented as mean ± standard error of mean (SEM) from a minimum of four experiments unless otherwise indicated. The cumulative probability of survival was analyzed by the Kaplan–Meier survival curve analysis, and the results were compared with the log-rank test followed by Bonferroni correction. A *p*-value of less than 0.05 was considered statistically significant.

## 3. Results

### 3.1. Chemical Properties of αGD2-IR700

SDS-PAGE and SEC were performed to evaluate the chemical properties of αGD2-IR700. SDS-PAGE showed that the position of the band of the IR700 fluorescence signal coincided with the position of the antibody band (Figure 1A). SEC analysis also confirmed that the absorption peak at 689 nm, which is the maximum absorption wavelength for IR700, was coincident with the monomer peak at 280 nm that eluted at 12.0 min for αGD2 antibody (Figure 1B). These results indicated that IR700 was successfully conjugated with the αGD2 antibody. Then, we quantified the number of IR700 conjugated with αGD2 antibody from absorption values at 280 nm and 689 nm in the UV-vis system. The number of IR700 bound to αGD2 antibody was 3.86 ± 0.22.

### 3.2. In Vitro NIR-PIT for GD2-Expressing Cancer Cell Lines

GD2 antigen is identical in mice and humans; therefore, we examined GD2 expression in three cancer cell lines from both species: EL4-luc (murine lymphoma), LAN-1 (human neuroblastoma), and T-98G (human glioblastoma). Of the three cell lines, LAN-1 uniformly expressed GD2 at a high level, and the other two cell lines showed partial expression (Figure 1C). Next, the cytotoxic effects of GD2-targeted NIR-PIT were investigated in vitro. Microscopic studies showed morphological changes in each cell line before and after in vitro NIR-PIT (Figure 1D) that were similar to NIR-PIT treated cells using other target molecules which we previously reported in detail [25]. Cultured cells incubated with αGD2-IR700 were irradiated with NIR light (16 J/cm^2^). Soon after NIR light irradiation, the cells began to swell and form blebs. After the GD2-targeted NIR-PIT, all three cell lines became PI-positive, indicating that the cell membrane’s integrity was lost. The cytotoxicity of GD2-targeted NIR-PIT was then quantified by flow cytometry. The cells were stained with PI after NIR-PIT and the dead cell percentage was calculated. Non-labeled and non-irradiated cells were used as controls. In all three cell lines, increased cell death was observed in a light dose-dependent manner (Figure 1E). LAN-1 cells were killed more than 95% by 8 J/cm^2^ irradiation, EL4-luc cells were killed more than 95% by 32 J/cm^2^ irradiation, and T98G cells were killed more than 60% by 64 J/cm^2^ irradiation. Therefore, LAN-1 is the most sensitive to GD2-targeted NIR-PIT, which is compatible with it possessing the highest expression of GD2. The anti-GD2 antibody used in this study is known to induce cell death by itself in GD2-positive cells in in vitro experiments [26]. This may be the reason for the reduced cell viability of labeling-only non-irradiated LAN-1 cells. Indeed, incubation with either unconjugated antibodies or αGD2-IR700 showed decreased cell viability (Appendix A), although GD2-targeted NIR-PIT further enhanced cytotoxicity.

### 3.3. Evaluation of αGD2-IR700 Uptake in a Mouse Tumor Model by In Vivo Fluorescence Imaging

To evaluate in vivo pharmacokinetic profiles of αGD2-IR700, an in vivo fluorescence imaging study was performed. Among the three cell lines, EL4-luc cell line was selected for the mouse tumor model because it could be evaluated with bioluminescence to detect early therapeutic effects of NIR-PIT [27]. In addition, EL4-luc cells produce a mouse syngeneic tumor model, making it possible to evaluate the immune response in mice. αGD2-IR700 was administered to EL4-luc tumor-bearing mice, and serial fluorescence images were obtained over the course of 6 days after injection (n = 7 mice per group) (Figure 2A). Accumulation of αGD2-IR700 in the tumor was detected one hour after administration of αGD2-IR700. The fluorescence intensity of αGD2-IR700 in the EL4-luc tumor reached its peak 9 h after αGD2-IR700 injection and gradually decreased thereafter. αGD2-IR700 was also taken up in the liver. The fluorescence intensity of αGD2-IR700 in the liver reached its peak 3 h after αGD2-IR700 injection (Figure 2B). The maximum TBR of αGD2-IR700 in tumor and liver was 612% and 331%, respectively. After reaching its peak, TBR remained at approximately the same level for several days (Figure 2C).

### 3.4. Evaluation of In Vivo Therapeutic Effect of GD2-Targeted NIR-PIT

Figure 3A shows a treatment regimen and imaging study protocol. For this experiment, EL4-luc tumor-bearing mice were randomly separated into the following three groups: Ctrl group (no treatment), ADC i.v. group, and NIR-PIT group. One day after injection of 100 μg of αGD2-IR700 via tail vein, the tumors of mice in the NIR-PIT group were exposed to NIR light (50 J/cm^2^). After NIR light exposure, the IR700 fluorescence signal in the tumor decreased (Figure 3B). Anti-GD2 antibody clone 14G2a has been reported to suppress the growth of GD2-positive tumors in vivo [28]. Consequently, in our study, the ADC i.v. group showed lower luciferase activity, slower tumor growth, and prolonged survival compared to the Ctrl group (Figure 3C–F). Tumor luciferase activity in the NIR-PIT group was significantly lower than in the other two groups (Figure 3C, D). Furthermore, when compared to the other two groups, tumor growth was significantly inhibited in the NIR-PIT group (Figure 3E), and significantly improved survival duration was observed in the NIR-PIT group (Figure 3F). No abnormal neurological findings were noted in all groups.

### 3.5. Histological Changes and Activated Antitumor Host Immunity after GD2-Targeted NIR-PIT

For histological analysis of tumors after GD2-targeted NIR-PIT, tumors were excised a day after NIR-PIT. Formalin-fixed paraffin-embedded sections of excised tumors were stained with hematoxylin and eosin. Although tumors in the Ctrl group and ADC i.v. group showed no obvious histological change, tumors in the NIR-PIT group showed diffuse necrosis and microhemorrhage. Next, we evaluated the activation of antitumor host immunity after GD2-targeted NIR-PIT. Regional lymph nodes (ipsilateral inguinal lymph nodes) were harvested 1 day after NIR-PIT and analyzed by flow cytometry. CD69 and CD25 were used as early and late activation markers of cytotoxic (CD3+CD8+CD45+) T cells. The percentage of CD25+ cells among CD8+ T cells within lymph nodes was significantly higher in the NIR-PIT group than in the other two groups, while there was no difference for CD69+ killer T cells (Figure 4B). The numbers of NK and NKT cells increased in the tumor-draining lymph nodes (Figure 4C). These results indicated antitumor host immunity was activated by GD2-targeted NIR-PIT.

## 4. Discussion

GD2 is a subtype of gangliosides (glycolipids) that is highly expressed mainly in tumors of neuroectodermal origin. In the present study, we demonstrated that GD2 is a promising target for NIR-PIT in GD2-expressing tumors. GD2-targeted NIR-PIT showed higher therapeutic efficacy compared to anti-GD2 monoclonal antibody administration alone both in vitro and in vivo.

In NIR-PIT, the photochemical changes in IR700 after irradiation of near-infrared light cause physical damage to the cell membrane by aggregating membrane-bound proteins [17]. To date, NIR-PIT has been developed to target a wide variety of transmembrane proteins such as EGFR, human epidermal growth factor receptor-2 (HER2), and prostate-specific membrane antigen (PSMA) [29,30]. Unlike transmembrane proteins, glycolipids are found only in the outer leaflet of the plasma membrane. Therefore, the physical forces exerted by glycolipid aggregation were posited to be relatively small, perhaps too small to disrupt cell membranes. However, this study showed that NIR-PIT targeting glycolipids caused physical damage to plasma membranes to cause cytotoxicity to a similar degree as NIR-PIT targeting transmembrane proteins. In addition, GD2-targeted NIR-PIT led to the activation of CD8-positive killer T cells, similar to other NIR-PITs targeting transmembrane proteins [31,32]. Increased numbers of NK cells and NKT cells were also observed in regional lymph nodes. These results suggest that both adaptive immunity and innate immunity were activated and that the results were similar to prior results using transmembrane proteins as the target for NIR-PIT.

This is the first report showing the effectiveness of NIR-PIT targeting a glycolipid; a non-protein target. With oncogenesis, conversion of glycolipid synthesis pathways occurs, resulting in changes in glycolipid composition [33,34]. As cancer-specific glycolipids, previous studies have reported the specific expression of Fuc-GM1 in small cell lung carcinoma and hepatocellular carcinoma, OAcGD3 in glioblastoma and lymphoblastic leukemia, and Globo H in breast cancer [35,36,37,38,39]. Therefore, this study has the potential to expand the number of potential targets for NIR-PIT.

Due to the insufficient therapeutic efficacy of anti-GD2 monoclonal antibody treatment alone, several novel approaches to anti-GD2 monoclonal antibody-based immunotherapy have been recently developed, including radiolabeled antibodies, immunotoxins, antibody-drug conjugates, and GD2-specific chimeric antigen receptors (CAR)-modified T cells [13]. Among these, GD2-specific CAR T cell therapy is the closest to clinical translation [40]. However, Richman et al. have reported that high-affinity GD2-specific CAR T cells induced severe neurological toxicity, including gait disturbance, inactivity, and seizure in mouse models [41]. GD2 antigen is identical in rodents and humans, and endogenously expressed antigens enable the analysis of on-tumor/off-target toxicity. In the present study, mice in the NIR-PIT groups did not show such neurological symptoms.

Recently, Battula and Ly et al. reported that GD2 is a novel cancer stem cell-specific cell surface marker in triple-negative breast cancer [42,43]. In preclinical studies, NIR-PIT targeting CD44 and CD133, which are known as cancer stem cell markers in breast cancer and glioblastoma, has been demonstrated to inhibit tumor growth [44,45]. Based on the results of these studies, GD2-targeted NIR-PIT might be effective in the treatment of triple-negative breast cancer. In another recent study, a possible synergistic effect of the combination of anti-GD2 and anti-CD47 antibodies has been reported in syngeneic mouse models of neuroblastoma and osteosarcoma [46]. CD47 is an innate immune checkpoint, the so-called ‘Do not eat me’ signal, and the efficacy of CD47-targeted NIR-PIT has been also reported [47]. Combination therapy of GD2-targeted NIR-PIT and CD47-targeted NIR-PIT will be a topic for future investigation.

While the results presented here are promising and have the potential for development, there are several limitations to this study. The first limitation is the use of an ectopic tumor model. Orthotopic tumor models are superior to subcutaneous (ectopic) tumor models in that the former can mimic the actual tumor microenvironment and drug distribution. Further investigation using orthotopic tumor models is needed. The second is the use of an aggressively proliferating cell line, EL4-luc cells. Although the use of EL4-luc cells enabled the analysis of host immune reactions induced by GD2-targeted NIR-PIT, the volume of EL4-luc tumors reached more than 2000 mm^3^ within 3 weeks, even in the NIR-PIT group. This rate of growth is atypical for most cancers. Fortunately, no abnormal neurological findings were observed during this period, but a more extended observation period would be ideal for future clinical application from a safety perspective.

In conclusion, although NIR-PIT has conventionally targeted transmembrane proteins, this study demonstrates that membrane glycolipids are another potential target. Recent studies have revealed many tumor-specific glycolipids and that NIR-PIT could be targeted to these and thereby applied to such cancers to improve therapeutic effects.

## Figures and Tables

**Figure 1 pharmaceutics-14-02037-f001:**
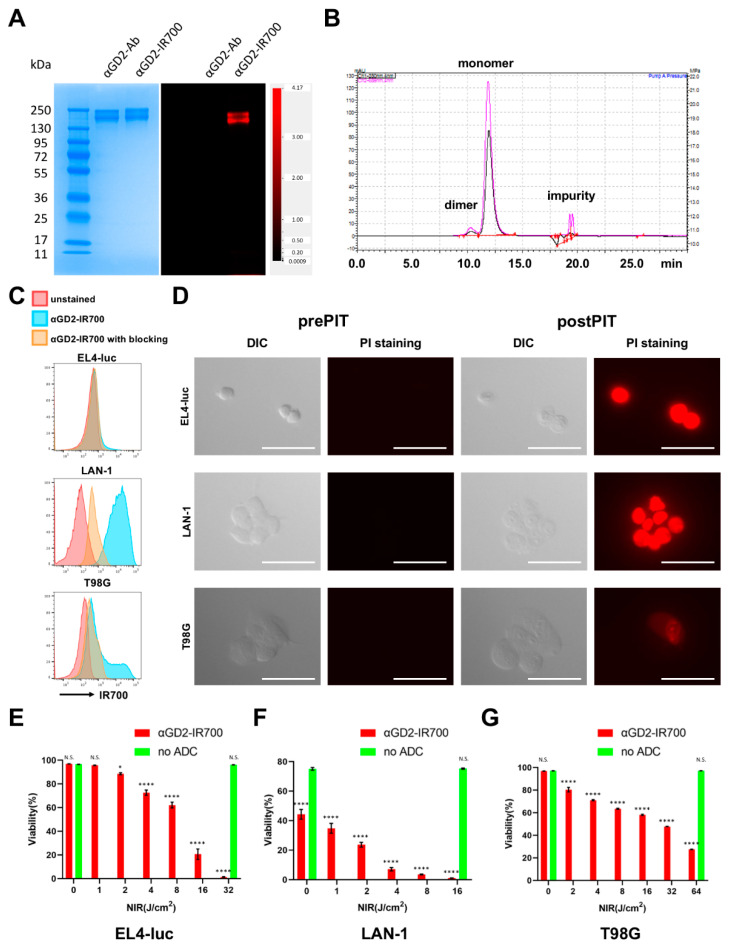
**Characterization of αGD2-IR700 and cytotoxic effects of in vitro GD2-targeted NIR-PIT**. (**A**) Validation of covalently bound IR700 to αGD2 antibody by SDS-PAGE (left: colloidal blue staining, right: IR700 channel). Unconjugated αGD2 antibody was used as a control. (**B**) SEC analysis of αGD2-IR700. The pink line points to the absorption at 690 nm, and the black line points to the absorption at 280 nm. (**C**) Expression analysis of GD2 in EL4-luc, LAN-1, and T98G cell lines using flow cytometry. Red, blue and orange histograms show unstained, αGD2-IR700 staining without blocking, and with blocking, respectively. (**D**) Differential interference contrast (DIC) images and fluorescence images before and after GD2-targeted NIR-PIT. Dead cells were stained with propidium iodide (PI). Scale bar = 50 μm. (**E**) Light-dose-dependent cell death in EL4-luc cells induced by GD2-targeted NIR-PIT. Cell viability was analyzed by flow cytometry after PI staining. Data are shown as mean ± SEM (n = 4, * *p* < 0.05, **** *p* < 0.0001 vs. no light exposure group, N.S. not significant). (**F**) Dose-dependent cell death in LAN-1 cells induced by GD2-targeted NIR-PIT. Cell viability was measured by PI staining. Data are shown as mean ± SEM (n = 4, **** *p* < 0.0001 vs. no light exposure group, N.S. not significant). (**G**) Dose-dependent cell death in T98G cells induced by GD2-targeted NIR-PIT. Cell viability was measured by PI staining. Data are shown as mean ± SEM (n = 4, **** *p* < 0.0001 vs. no light exposure group, N.S. not significant).

**Figure 2 pharmaceutics-14-02037-f002:**
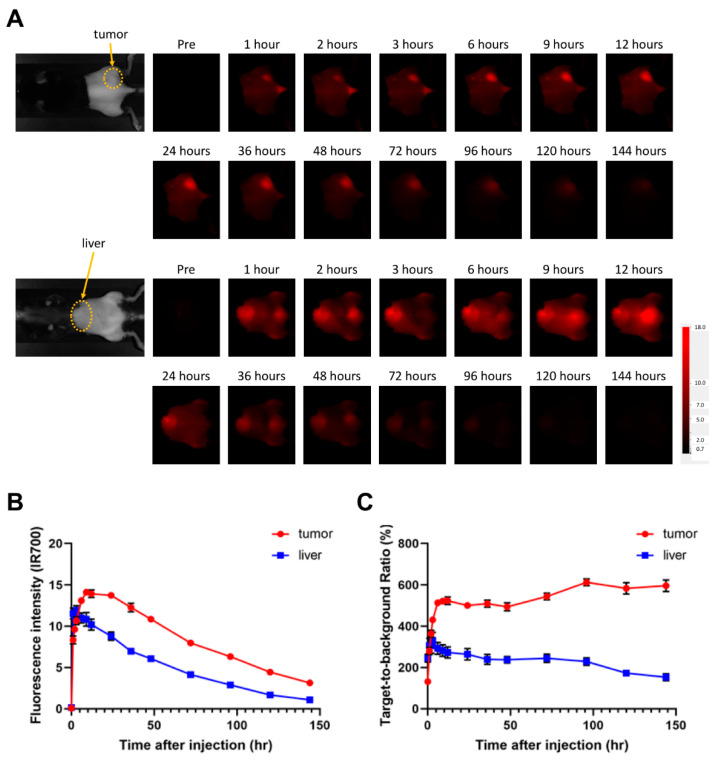
**Biodistribution of αGD2-IR700 in EL4-luc tumor bearing mice.** (**A**) In vivo serial fluorescence images of the EL4-luc tumor-bearing mice. Mice were intravenously administered with αGD2-IR700, and IR700 fluorescence images were obtained at the time points indicated. (**B**) Changes in IR700 fluorescence intensity in tumor and liver. Data are shown as mean ± SEM. Data were obtained from seven animals at each time point. (**C**) Changes in target-to-background ratios in tumor and liver. Data are shown as mean ± SEM. Data were obtained from seven animals at each time point.

**Figure 3 pharmaceutics-14-02037-f003:**
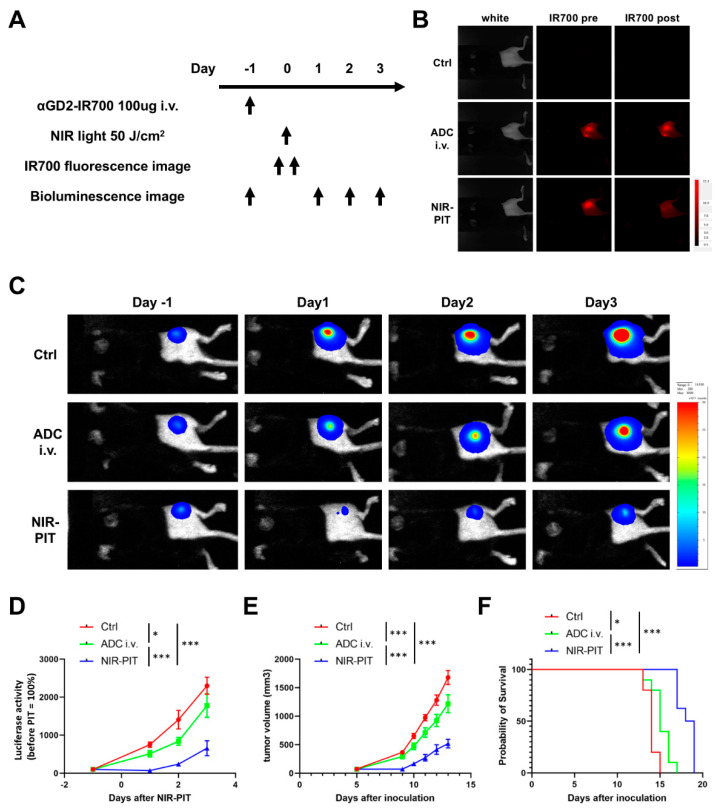
**In vivo tumor response of GD2-targeted NIR-PIT.** (**A**) Treatment regimen and imaging study protocol. (**B**) Representative IR700 fluorescence images of mice with EL4-luc tumor before and after GD2-targeted NIR-PIT. (**C**) Representative bioluminescence images of mice with EL4-luc tumor. (**D**) Quantitative analysis of changes in luciferase activity in EL4-luc tumors (values before treatment were set to 100%; n = 10 animals per group; two-way repeated measures ANOVA followed by Tukey’s multiple comparison test; * *p* < 0.05, *** *p* < 0.001). (**E**) Tumor growth curves (n = 10 animals per group; two-way repeated measures ANOVA followed by Tukey’s multiple comparison test; *** *p* < 0.001). (**F**) Survival curves (n = 10; log-rank test followed by Bonferroni correction; * *p* < 0.05, *** *p* < 0.001).

**Figure 4 pharmaceutics-14-02037-f004:**
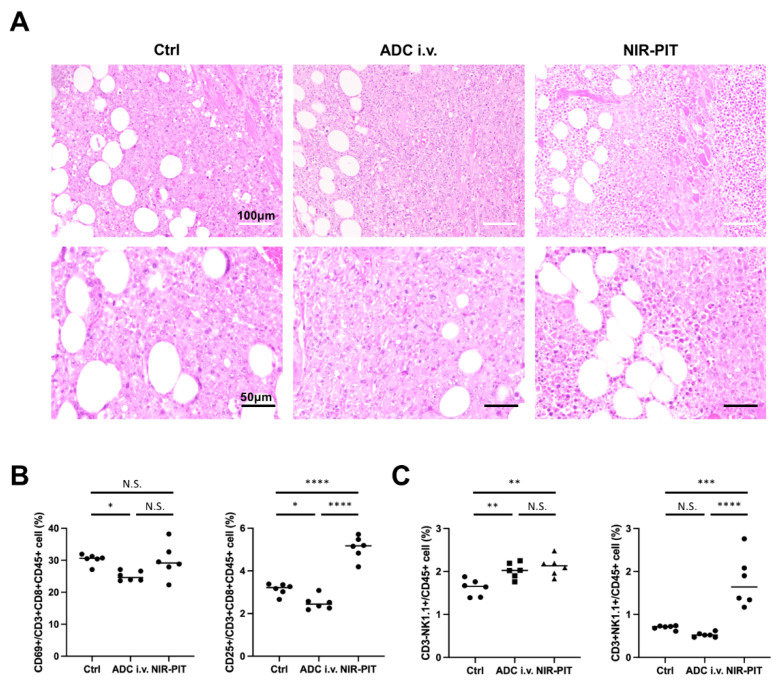
**Pathological analysis of EL4-luc tumor after GD2-targeted NIR-PIT.** (**A**) Hematoxylin and eosin staining of tumors resected one day after NIR-PIT (White scale bars, 100 μm; black scale bars, 50 μm). (**B**,**C**) Immune cell response after GD2-targeted NIR-PIT was analyzed by flow cytometry 1 day after NIR-PIT. (**B**) Activation marker expression in CD8+ killer T cells in the regional lymph nodes. (n = 6; one-way ANOVA followed by Tukey’s test; * *p* < 0.05, **** *p* < 0.0001; N.S., not significant) (**C**) Number of NK and NKT cells in the regional lymph nodes. (n = 6; one-way ANOVA followed by Tukey’s test; ** *p* < 0.01, *** *p* < 0.001, **** *p* < 0.0001; N.S., not significant).

## Data Availability

Data are available when reasonably requested to the corresponding author.

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
