# Peer review of "Disialoganglioside GD2-Targeted Near-Infrared Photoimmunotherapy (NIR-PIT) in Tumors of Neuroectodermal Origin"

_pharmaceutics, 2022, doi:10.3390/pharmaceutics14102037_

Round 1
Reviewer 1 Report
The paper is well written and deal with important topic of photoimmuno therapy. The research is well presented and design and the statements are supported by the results. Only some typos are present. Please add information how many mice where used in each group, how they were kept, how did you collect samples and so on.
Author Response
Reviewer #1 Comments and Suggestions for Authors
The paper is well written and deal with important topic of photoimmuno therapy. The research is well presented and design and the statements are supported by the results. Only some typos are present. Please add information how many mice where used in each group, how they were kept, how did you collect samples and so on.
We thank the reviewer for the careful review and the positive comments.
According to the reviewer’s comments, the following information were added in the materials and methods section.
thirty EL4-luc tumor-bearing mice were randomly divided into the following three groups (p5, l177)
Mice were maintained under a 12-hour light-dark cycle. (p4, l153-154)
One day after NIR-PIT, tumors were excised for histological studies. (p5, l191)
We corrected some typos.
P5, l171: Backround -> Background
P15, l395: markes -> markers

Reviewer 2 Report
To address the less optimistic outcome of anti-GD2 on patients, similar solution (antibody-photoabsorber conjugates) as other antibodies (such as anti-EGFR) used in clinical or preclinical trials, authors proposed anti-GD2 antibody-IR700 strategy to potentiate therapeutic effect of anti-GD2 on cancer therapy, which may provide the valuable guidance for clinical application of upgraded anti-GD2, although the novelty of the work is not very fantastic. Some suggestions supplied for improving your manuscript.
1. For better testify the targeting site on the cell membrane, nucleus, and cell cytosol should be dyed separately, and more fine cell structure changes before/after irradiation should be supplied to replace Fig 1D which is not very clear for comparison the fine structure changes.
2. The analysis of immune cells in spleen should be also taken into consideration for better understanding the activation of host immunity. The changes on tumor size and mice weight should be supplied.
3. The therapy results on tumor laden mice showed that the improve outcome is not very exciting, which just exhibit around 3-5 days longer survival time comparing the control or ADC groups, Pls. provide the more possible reasons for this phenomenon, as well as the direction of effort in the near future on this research proposal.
Author Response
Reviewer #2 Comments and Suggestions for Authors
To address the less optimistic outcome of anti-GD2 on patients, similar solution (antibody-photoabsorber conjugates) as other antibodies (such as anti-EGFR) used in clinical or preclinical trials, authors proposed anti-GD2 antibody-IR700 strategy to potentiate therapeutic effect of anti-GD2 on cancer therapy, which may provide the valuable guidance for clinical application of upgraded anti-GD2, although the novelty of the work is not very fantastic. Some suggestions supplied for improving your manuscript.
- For better testify the targeting site on the cell membrane, nucleus, and cell cytosol should be dyed separately, and more fine cell structure changes before/after irradiation should be supplied to replace Fig 1D which is not very clear for comparison the fine structure changes.
Thank you for your comment. We have already reported the detailed spatial and temporal morphological changes before, during, and after NIR-PIT, using three-dimensional low coherent quantitative phase microscopy (3D LC-QPM), (Ogata et al. Oncotarget. 2017; 8(61): 104295-104302). 3D LC-QPM showed region-specific cell membrane rupture occurring first on the distal free edge of the cell near the site of adhesion, in a process that was independent of cell shape. These results indicated that the target site of NIR-PIT was cell membrane. We have added this discussion with the reference.
- The analysis of immune cells in spleen should be also taken into consideration for better understanding the activation of host immunity. The changes on tumor size and mice weight should be supplied.
Thank you for your helpful suggestions. Analysis of immune cells in the spleen would be helpful for better understanding of host anti-tumor immunity activation, however we did not take such data because we intended to evaluate if IR700 conjugated anti-GD2 Ab, a therapeutic antibody by itself, could work as a NIR-PIT agent for strengthen the therapeutic effects. As for the changes in tumor size, we have already shown it in Figure 3E. Although the exact changes in the mice's weight were not recorded, the appearance of the mice did not differ among the three groups.
- The therapy results on tumor laden mice showed that the improve outcome is not very exciting, which just exhibit around 3-5 days longer survival time comparing the control or ADC groups, Pls. provide the more possible reasons for this phenomenon, as well as the direction of effort in the near future on this research proposal.
EL4-luc cells used in this in vivo experiment are very fast-proliferating cells, and the mice in the control group died around only 2 weeks after inoculation. Three to five days prolonged survival by GD2-targeted NIR-PIT was approximately 30% longer survival that was significantly different from control groups. We might try this in future study when a better tumor model is available. In addition, as we already addressed in the Discussion section (p15, l397-402), combination therapy with CD47-targeted NIR-PIT has the potential to further prolong survival.
